

# DNA-based prediction of external ear morphology in the Chinese population: an exploratory study

Xindi Wang[1,*], Zibo Zhao[2,*], Jingting Wu[3], Yuan Li[3], Yufei Yang[1], Bo Liu[1], Chengye Zhou[1], Chuanxu Wang[1], Xiaogang Chen[3] and Feng Song[1]

[1] Department of Forensic Genetics, Sichuan University, Chengdu, China
[2] Department of Artificial Intelligence, College of Blockchain Industry, Chengdu University of Information Technology, Chengdu, China
[3] Department of Forensic Pathology, Sichuan University, Chengdu, China
[*] These authors contributed equally to this work.

## ABSTRACT

**Background**. The external human ear is a polymorphic and polygenic structure with individual uniqueness, making it a valuable target in forensic DNA phenotyping (FDP) studies. Previous genome-wide association studies have identified multiple genetic loci associated with variation in ear characteristics. However, research focused on predicting ear morphology within the context of FDP remains limited. This study aimed to develop DNA-based predictive models for external ear morphology in the Chinese population.
**Methods**. Digital photographs of 675 volunteers were used to score 13 ear phenotypes, each categorized into three levels. Multinomial logistic regression (MLR) was applied for genetic association analysis. Five predictive models—MLR, support vector machines, random forest, AdaBoost, and k-nearest neighbors—were developed and evaluated using 10-fold cross-validation.
**Results**. Genetic association analysis identified several influential single-nucleotide polymorphism (SNPs) for each ear phenotype. Among the five models, AdaBoost and MLR demonstrated superior performance, achieving area under the curve (AUC) values above 0.7 for predicting absent tragus cases (level_0). To simplify classification, binary models incorporating genetic interactions were constructed for absent tragus cases. Specifically, the AdaBoost model achieved an AUC of 0.74, while the binary logistic regression (BLR) model reached an AUC of 0.72.
**Conclusions**. These findings highlight the potential forensic application of genetic markers in predicting ear morphology within the Chinese population, contributing to the advancement of FDP research and practice.

# INTRODUCTION

Forensic DNA phenotyping (FDP) refers to the prediction of a person's externally visible characteristics (EVCs) based on DNA extracted from human biological samples collected at a crime scene (*Dabas et al., 2022*; *Kayser et al., 2023*). By predicting biological traits

Corresponding authors
Xiaogang Chen, gchan76@163.com
Feng Song, fengsong9@163.com

such as appearance, biogeographic ancestry, and age, FDP plays a vital role in criminal investigations, disaster victim identification, and archaeological research (*Alshehhi et al., 2023*). For instance, in a 2009 cold case resolution, FDP analysis of blood evidence enabled phenotypic reconstruction of a suspect's skin color, eye color, and freckling, supporting an arrest in 2016 (*Kennedy, 2022*). Since 2011, DNA panels for predicting eye, hair, and skin color have been established, such as IrisPlex (*Walsh et al., 2011*), HIrisPlex (*Walsh et al., 2013*), and HIrisPlex-S (*Chaitanya et al., 2018*). Additionally, more DNA markers have been discovered for physical appearance, including eyebrow color (*Peng et al., 2019*), freckles (*Hernando et al., 2018*; *Kukla-Bartoszek et al., 2019*), hair shape (*Liu et al., 2018*), baldness (*Liu et al., 2016*; *Hagenaars et al., 2017*), height (*Liu et al., 2019*), and facial features (*Liu et al., 2021*; *Zhang et al., 2022*). However, research on predicting external ear morphology remains in its early stages.

The external ear, as an integral part of facial features, plays a significant role in shaping an individual's overall appearance (*Guyomarc'h & Stephan, 2012*; *Hiware et al., 2024*). Unlike other facial features, ear morphology remains relatively stable, unaffected by emotional states or the use of face masks, thus serving as a crucial target in FDP studies (*Hiware et al., 2024*; *Benzaoui et al., 2023*). This stability, combined with its high degree of variability among individuals, makes the external ear a valuable tool in forensic investigations for individual identification (*Rani et al., 2020*; *Rani, Krishan & Kanchan, 2022*; *Fakorede et al., 2021*; *Baroniya, Harshey & Srivastava, 2021*). Moreover, variations of ear morphology are influenced by factors such as age (*Sforza et al., 2009*), sex (*Rani et al., 2020*; *Rani et al., 2021*; *Krishan, Kanchan & Thakur, 2019*), ethnicity (*Verma et al., 2016*), and bilateral asymmetry (*Rani et al., 2020*; *Rani et al., 2021*; *Krishan, Kanchan & Thakur, 2019*), with even subtle differences observed between twins (*Zulkifli, Yusof & Rashid, 2014*).

The variability in ear morphology is ultimately shaped by the interaction between genetic factors and environmental conditions (*Rani, Krishan & Kanchan, 2022*; *Samuel & Prainsack, 2019*). Genome-Wide Association Studies (GWAS) have identified several genetic loci associated with ear morphology across populations, providing a solid research foundation for FDP (*Adhikari et al., 2015*; *Shaffer et al., 2017*; *Wang et al., 2022*; *Li et al., 2023*). In an FDP study on the Pakistani population, *Noreen et al. (2023)* demonstrated the feasibility of predicting 11 ear traits using 20 single-nucleotide polymorphism (SNPs), achieving moderate to good prediction accuracy. Their finding has preliminarily highlighted the potential for DNA-based prediction of ear morphology. However, this study utilized a relatively small sample size of 300 individuals, limiting the generalizability of its conclusions. Moreover, given the genetic disparity across populations, the predictive performance of these previously reported loci requires additional validation in the Chinese cohort.

Currently, machine learning (ML) has emerged as a powerful tool for addressing classification challenges across various domains (*Faragalli et al., 2025*; *Kapoor, Sharma & Sharma, 2024*; *Tian, Tian & Zhao, 2024*; *Jung et al., 2024*). The ML algorithms are trained on datasets to estimate model parameters and make informed decisions (*Katsara et al., 2021*). Most FDP studies utilized logistic regression (LR) algorithms for phenotypic prediction (*Chaitanya et al., 2018*; *Hernando et al., 2018*; *Kukla-Bartoszek et al., 2019*).
However, complex phenotypes exhibit intricate non-linear relationships and high-dimensional data patterns, making it challenging for a single model to capture these characteristics effectively. Therefore, a comparative analysis of multiple ML models is essential to achieve optimal predictive performance.

To address these issues, this study explored genetic markers associated with ear phenotypes and investigated gene-gene interactions in a cohort of 675 Chinese individuals. Multiple DNA-based prediction models were established for both three-category and binary ear phenotypes. Using cross-validation methods, our work established and evaluated the predictive efficacy of DNA-based models for ear morphology. By developing an analytical framework for ear morphology, this study is expected to extend FDP applications to a novel phenotypic domain, providing fundamental insights to EVC reconstruction in forensic casework.

## MATERIALS & METHODS

### Samples collection

A total of 675 Chinese volunteers, aged 18–93 years (429 males and 246 females), participated in this study. DNA was extracted from oral swabs using the phenol–chloroform isoamyl alcohol (*Butler, 2012*) and quantified by NanodropTM 2000 spectrophotometer (Thermo Scientific, Waltham, MA, USA). Five digital photographs of the head were taken at the Frankfort horizontal plane using a Canon 80D camera: the left side (90°), the left angle (45°), a frontal view (0°), the right angle (45°), and the right side (90°) (*Adhikari et al., 2015*). This study was approved by the Ethics Committee at Sichuan University (k2020032). This work was carried out in accordance with the principles of the Declaration of Helsinki, and all donors gave written informed consent.

### Human ear phenotypes

Right side, right angle, and frontal photographs were used to assess 13 ear traits on a three-point categorical scale (level_0, level_1, and level_2). The ear traits evaluated included ear protrusion, lobe attachment, lobe size, tragus size, antitragus size, intertragic incisure, superior helix rolling, posterior helix rolling, folding of the antihelix, antihelix curvature, crus helix expression, superior crus of antihelix expression, and Darwin's tubercle (Fig. 1). These traits and the scoring standards were in accordance with previous studies (*Rani, Krishan & Kanchan, 2022*; *Adhikari et al., 2015*; *Verma, Bhawana & Vikas, 2014*). Intraclass correlation coefficients (ICCs) were calculated to assess inter- and intra-observer reliability (*Shrout & Fleiss, 1979*). The detailed methods were described in Table S1. All photographs of the volunteers were scored by the same rater (X.W.). The frequency distribution of the three categories for each ear trait was presented in Fig. S1. The Spearman's rank correlation coefficients were calculated to assess the correlations among the various ear traits.

### Candidate marker selection

The SNPs reported by large-scale GWASs that exhibited statistically significant associations with ear morphology were initially selected (*Adhikari et al., 2015*; *Shaffer et al., 2017*; *Wang et al., 2022*; *Li et al., 2023*). Duplicate variants identified across different studies

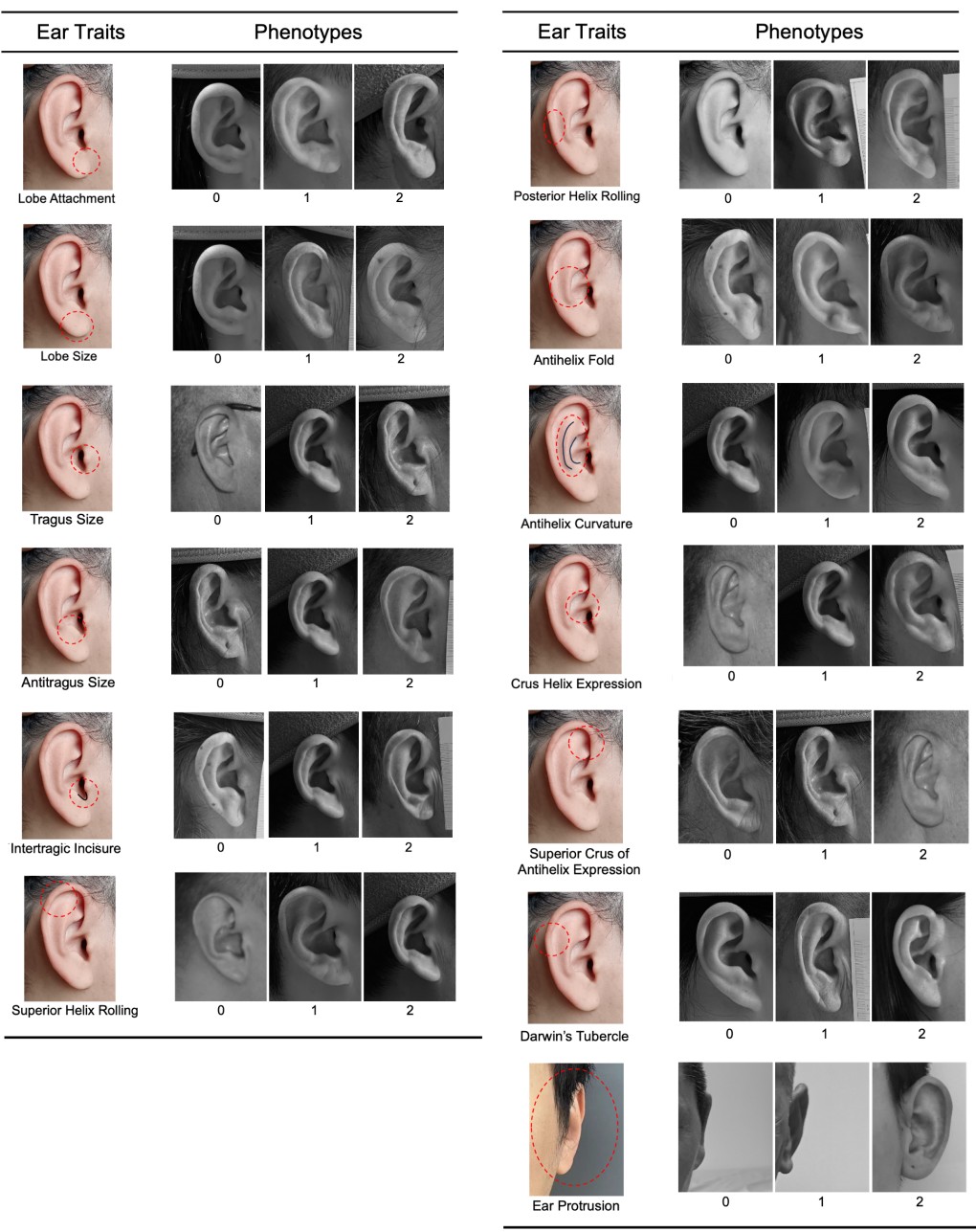

**Figure 1** **Categorical phenotyping of examined ear traits.** Each ear trait was scored into three levels: level_0, level_1, and level_2.

were carefully eliminated. We then refined our selection by filtering out markers with a minor allele frequency (MAF) below 0.05, based on population-specific data from the Beijing Han Chinese (CHB) and Southern Han Chinese (CHS) of the 1000 Genomes Project Phase 3 (https://www.internationalgenome.org). LDlink online software (https://ldlink.nih.gov/?tab=home) was utilized to exclude SNPs that were in high linkage disequilibrium (LD) with each other in the CHB and CHS, based on the GRCh38.p14

reference genome (*Machiela & Chanock, 2015*). Specifically, an LD threshold of $r^2 < 0.5$ was set to ensure that our final list of SNPs was as independent as possible.

## Multiplex SNaPshot assay

Primers for amplification and single-base extension (SBE) were designed following the operating instructions of the PyroMark Assay Design 2.0 software. To validate the specificity of the polymerase chain reaction (PCR) primers, *in-silico* analysis was performed using the UCSC genome browser (*Hendling & Barišić, 2019*). Meanwhile, both PCR and SBE primers were analyzed using Autodimer to detect potential hairpin and primer dimer formations (*Vallone & Butler, 2004*). The PCR primers were further examined according to the results of polyacrylamide gel electrophoresis (PAGE), and the SBE primers were examined according to the single-site SBE assay.

The multiplex PCR was performed in a five-µl reaction volume containing two µl PyroMark PCR Master Mix (Qiagen, Hilden, Germany), one µl of mixed PCR primers (Table 1), two ng of DNA, and RNase-free water. The PCR amplification was carried out under the following conditions: initial denaturing at 95 °C for 15 min, 20 cycles at 94 °C for 30s, 65 °C for 90s (drop down 0.5 °C each cycle), 72 °C for 30s, 19 cycles at 94 °C for 30s, 55 °C for 90s, 72 °C for 30s, followed by 72 °C for 10 min. The following PCR procedures were carried out as described in *Jiang et al. (2023)*. Specifically, two µl shrimp alkaline phosphatase (SAP) and one µl of exonuclease I were added to the PCR products, incubating at 37 °C for 1 h to eliminate the single-strand primers and dNTP, followed by heat inactivation at 80 °C for 10 min to eliminate the enzyme. Then, the mini-sequencing was performed in five µl reactions containing one µl of the PCR products dealt with digestion, one µl of mixed extension primers (Table 1), 1.2 µl SNaPshot Multiplex Ready Reaction Mix, and 1.8 µl RNase-free water. It was conducted under 26 cycles of 96 °C for 10 s, 50 °C for 5 s, and 60 °C for 30 s. To remove the remnant ddNTP, one µl SAP was then added to the product by incubation at 37 °C for 1 h, followed by heat inactivation at 80 °C for 10 min.

Purified products (one µL) were combined with nine µL of HiDi™ Formamide sizing standard mixture. This sizing standard mixture was prepared by adding three µL GeneScan™ 120 LIZ® Size Standard to one mL HiDi™ Formamide. Electrophoresis was performed on an ABI 3130xl Genetic Analyzer (Thermo Fisher Scientific, Waltham, MA, USA) with 36 cm capillary arrays and POP-7™ polymer. The specific reaction conditions for 3,130 were as follows. Samples were injected at 2.5 kV for 10s, and electrophoresed at 13 kV for 600 s at a run temperature of 60 °C. Raw data were analyzed in GeneMapper® ID v3.2 software (Applied Biosystems, Foster City, CA, USA) with a peak detection threshold of 100 relative fluorescence units (RFU).

## Association analysis

Allele frequencies for each SNP were calculated based on our dataset of the Chinese population. All genotype data were tested for Hardy-Weinberg (HW) equilibrium and LD analysis using the online software SHEsis (http://analysis.bio-x.cn/myAnalysis.php) (*Shi & He, 2006*).

**Table 1 Locations and primer information of SNP Markers in the SNaPshot assays.**

| Assay position | SNP ID[a] | Chr Number and Region[a] | Nearest Gene[a] | PCR Primer (5′–3′) | Product size (bp) | SBE prime (5′–3′) | Primer Concentration (μM) |
|---|---|---|---|---|---|---|---|
| Plex1_1 | rs3827760 | 2, missense | EDAR | F: CCCAATCTCATCCCTCTTCA; R: CAGCTCCACGTACAACTCTGA | 88 | GCCTCCTC CCCCGCCA CGTTTTCAC A | PCR: 0.2 SBE: 0.2 |
| Plex1_2 | rs17023457 | 1, intronic | CART1 | F: AAAAGGCATGAAAAATGATACCG; R: TGTGTTTTTGGTTAGGAACTGAAG | 82 | GCCTCCTC CCCTCCCC CGACCACT AACTAATC AACA | PCR: 0.2 SBE: 0.2 |
| Plex1_3 | rs74030209 | 16, intronic | ZFHX3 | F: TCTACTCCCCAACACAATACCC; R: AGTCTTCTGCATGTGGGAACTTT | 168 | GCCTCCTC CCCTCCCCT CCCCTCCTT ATAATGGA TACATGC | PCR: 0.8 SBE: 0.9 |
| Plex1_4 | rs6802174 | 3, intronic | MRPS22 | F: ACCCACTGGGCAGTAGCAGA; R: CTTTCACGCACAGGGATTGCT | 158 | GCCTCCTC CCCTCCCCT CCCCTCCC CTCCGAGG GATCGGTA TTGTA | PCR: 0.8 SBE: 1.1 |
| Plex1_5 | rs10198822 | 2, intronic | SP5 | F: TCTGAAGCTGCAGTTCCACTCC; R: TTTTTCCCTCCTTGTATCAGTCC | 129 | GCCTCCTC CCCTCCCCT CCCCTCCC CTCCCCTCC CCTGTCCG TTTGGGAT T | PCR: 0.2 SBE: 0.2 |
| Plex2_1 | rs6699106 | 1, intergenic | TBX15 | F: GTCCTAGGCTTTTCATTGATCAGA; R: ACATACAACCTGCCAAGACTGAA | 115 | GCCTCCCC TGAACAGC TCAATAAT | PCR: 0.2 SBE: 0.2 |
| Plex2_2 | rs1948400 | 3, intronic | MRPS22 | F: TTTACAGGTAGGGAGGCTGAGT; R: ACAATCAGAAAAGTGGGACAGTG | 87 | GCCTCCTCC CCTCCTGAG TGGGGACAG AG | PCR: 0.6 SBE: 0.8 |
| Plex2_3 | rs17034666 | 2, intronic | EDAR | F: GACCTGGCCGGGAAGATAA; R: CTTGCCCAAAGTTGCATAGCT | 197 | GCCTCCTCC CCTCCCCTC CCCTCTCCCT GAGGGAAGC | PCR: 1.0 SBE: 1.2 |
| Plex2_4 | rs7812632 | 8, intronic | HAS2-AS1 | F: ACCAAGGAATTTGGCAAAGACT; R: TACTGTCTGGAAGGGCTAATGACT | 215 | GCCTCCTCC CCTCCCCTC CCCTCCTCA AGGTCTTTG TGTCTA | PCR: 1.0 SBE: 1.0 |
| Plex3_1 | rs62169501 | 2, intronic | SP5 | F: CCAAGTTAGCCTGCTGTAGTTTC; R: CTGACCCAAAAGTCTTGTACCTTC | 60 | GCCTCCCCT TCTGCCAAT ATCC | PCR: 0.2 SBE: 0.4 |

**Table 1** (*continued*)

| Assay position | SNP ID[a] | Chr Number and Region[a] | Nearest Gene[a] | PCR Primer (5′–3′) | Product size (bp) | SBE prime (5′–3′) | Primer Concentration (μM) |
|---|---|---|---|---|---|---|---|
| Plex3_2 | rs3789101 | 2, intronic | *ACOXL* | F: CCCTCCTGCTTAAATGTCGTATC R: AGGGTCAAGTGCTGTTGAATCA | 174 | GCCTCCTCC CTTTTTGAC AACTTTCTCT CT | PCR: 0.2 SBE: 0.2 |
| Plex3_3 | rs263156 | 6, intronic | *LOC153910* | F: GTGATGCCCAAGCACTAAGTT R: CCTGGGCAGATTCTTGCTC | 76 | GCCTCCTC CCCTCCCC TCCTACCC TATCATTC CACC | PCR: 0.2 SBE: 0.2 |
| Plex3_4 | rs1960918 | 4, intronic | *LRBA* | F: AGGTTTGCCTGAGATAATTGAGTG R: GATGCAAATTTCAGGGATTTTGTT | 188 | GCCTCCTC CCCTCCCC TCCCTCCT TGAGTGAA TCTCGGTA A | PCR: 0.8 SBE: 0.8 |
| Plex3_5 | rs7771119 | 6, intergenic | *LOC153910* | F: CTCTCCTGTTTCAACGTTTTATCC R: AACTTGTTGCGGGCTTGG | 73 | GCCTCCTC CCCTCCCC TCCCCTCC CCTCCCCG CTCTATGTT GCCTCTTT | PCR: 0.2 SBE: 0.2 |
| Plex3_6 | rs1619249 | 18, intergenic | *LOC100287225* | F: CGGGGTTTTCACTTTATTAGCCAG R: GGGCGTGGTGGACTTTACATTTAC | 121 | TTTTTTTTT TTTTTTTTT TTTTTTTTT TTTTTTTTT TTTTTGGG CGGATAGG AGGC | PCR: 0.4 SBE: 0.8 |

**Notes.**

[a]Information based on the GRCh38 reference genome.

SNP, single nucleotide polymorphism; Chr, chromosome; PCR, polymerase chain reaction; SBE, single-base extension.

DNA variants with an MAF higher than 0.05 were tested for their association with each ear phenotype on the entire dataset of 675 samples using multinomial LR (MLR) analysis. Allelic odds ratios (ORs) with 95% confidence intervals and respective $p$-values were determined for minor alleles categorized in an additive manner. The effect of SNPs on the respective phenotypes was measured by the ORs (*Noreen et al., 2023*). $P$-value < 0.05 was considered nominally significant. The association analyses were conducted in R version 4.4.1 using the 'nnet' package.

**Prediction model**

A comparative analysis was conducted to assess the efficacy of five predictive models for three-class classification: MLR, SVM, RF, AdaBoost, and k-nearest neighbors (KNN). The SVM model was implemented with a radial basis function kernel and a regularization parameter set to 1.0, with probability estimates enabled. The RF classifier was constructed with 100 decision trees. The KNN model employed three nearest neighbors, while AdaBoost model was configured with 100 estimators and utilized the SAMME algorithm. These parameter selections were intended to optimize each model's performance in the multi-class

classification setting. Ten-fold cross-validation was employed to guard against arbitrary partitions of the dataset. Specifically, the dataset was randomly split into ten equal subsets. During each fold of the cross-validation process, nine of these subsets were designated as the training set, while the remaining subset served as the testing set. For each category, positive predictive value (PPV), negative predictive value (NPV), sensitivity, specificity, and area under the curve (AUC) values were calculated based on the average performance across the ten testing sets obtained from the iterations.

Specific categories of ear traits, exhibiting good predictive performance in their respective three-class classification tasks, were selected. To simplify the classification process, binary prediction models were developed for each selected category. Additionally, interactions between genetic variants were examined using the MDR 3.0.2 software. The multifactor dimensionality reduction (MDR) method is a powerful strategy for detecting and interpreting statistical locus-locus epistasis (*Moore et al., 2006*; *Moore, 2004*). As a data mining technique, it is used to detect and characterize non-linear relationships between variables (*Moore & Williams, 2009*; *Ritchie et al., 2001*). The dendrogram graphs provided by MDR illustrate the presence, strength, and nature of epistatic effects (*Moore et al., 2006*). Furthermore, the genetic interactions were tested by including interactions into our binary prediction models. The predictive workflow was illustrated in Fig. S2. Alternative thresholds of probability for the phenotype prediction were tested, ranging from $p > 0.5$ to $p > 0.85$ with a 0.05 interval. Prediction modelling was carried out using Python version 3.7.

## RESULTS

### Qualitative ear phenotypes

The ICCs ranged from 0.47 to 0.97, indicating moderate to good intra-rater and inter-rater consistency (Table S1). The scores for ear traits examined in the Chinese population showed a weak correlation between them (Table S2), with Spearman correlation coefficients all below 0.3. Specifically, there was a correlation between antihelix fold and superior crus of antihelix expression ($r = 0.218$). In contrast, lobe size exhibited a significant negative correlation with the ear protrusion ($r = -0.207$).

Five ear traits exhibited weak but statistically significant correlations with sex: superior helix rolling ($r = 0.343$), lobe size ($r = 0.219$), ear protrusion ($r = -0.209$), lobe attachment ($r = 0.127$), and posterior helix rolling ($r = -0.178$). Separately, age was negatively correlated with the scores of four ear traits: antitragus size ($r = -0.137$), superior helix rolling ($r = -0.136$), antihelix curvature ($r = -0.126$), and lobe attachment ($r = -0.124$) (Table S3).

### Candidate markers selection and multiplex SNaPshot assay

A list of 45 SNPs was initially selected following an exhaustive review of the literature on ear morphology. MAF screening and LD analysis were then applied to refine our selection, retaining 23 SNPs. However, designing feasible primers for six SNPs was challenging, including rs12695694, rs9496426, rs10211400, rs1602631, rs57788627, and rs62169502. Incorporating SNP rs2742261 into the multiplex system with other SNPs also posed

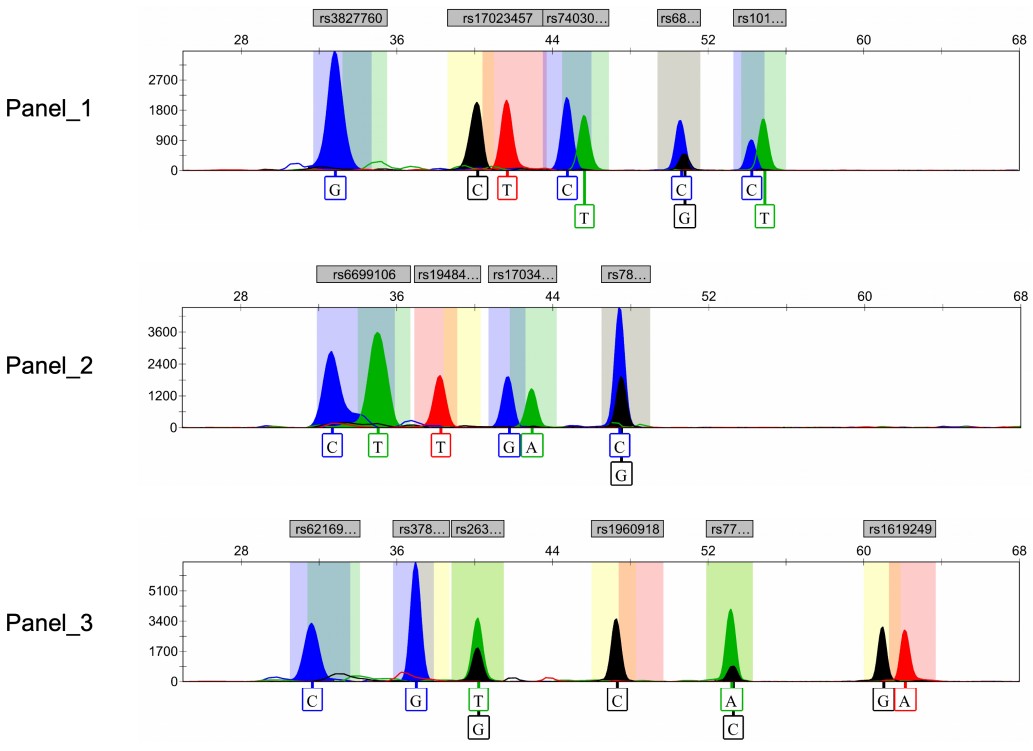

**Figure 2 Electropherograms of three SNaPshot multiplex assays containing 15 SNP markers.** SNP, single nucleotide polymorphism.

difficulties. Furthermore, upon examination of our dataset, we found that the SNP rs10824309 was monomorphic and thus excluded from the multiplex assays.

Ultimately, after rigorous screening and examination, a total of 15 genetic markers (rs10198822, rs6802174, rs74030209, rs17023457, rs17034666, rs3827760, rs7812632, rs1948400, rs6699106, rs263156, rs3789101, rs7771119, rs62169501, rs1619249, rs1960918) were included in the development of the final multiplex SNaPshot assays (Fig. 2). All peaks were detected with a 100 RFU threshold and all blood samples were successfully genotyped, underscoring the successful establishment of the robust and reliable SNaPshot system.

## Association analysis

The allele frequencies were shown in Table S4, and deviations from HWE were noted for eight SNPs (Table S5). Details of LD analysis are shown in Table S6, indicating that all the SNPs were in linkage equilibrium.

All SNP markers included in the association analysis, except for rs17034666, were found to be associated with ear traits in the Chinese cohort. Among these SNPs, rs6802174 exhibited significant associations with six ear traits: ear protrusion, lobe attachment, tragus size, antihelix fold, antihelix curvature, and Darwin's tubercle. Subsequently, rs74030209 was significantly associated with five ear traits: ear protrusion, lobe attachment, lobe size, antihelix curvature, and crus helix expression. Similarly, rs1948400 demonstrated significant associations with five ear traits, including ear protrusion, lobe attachment,

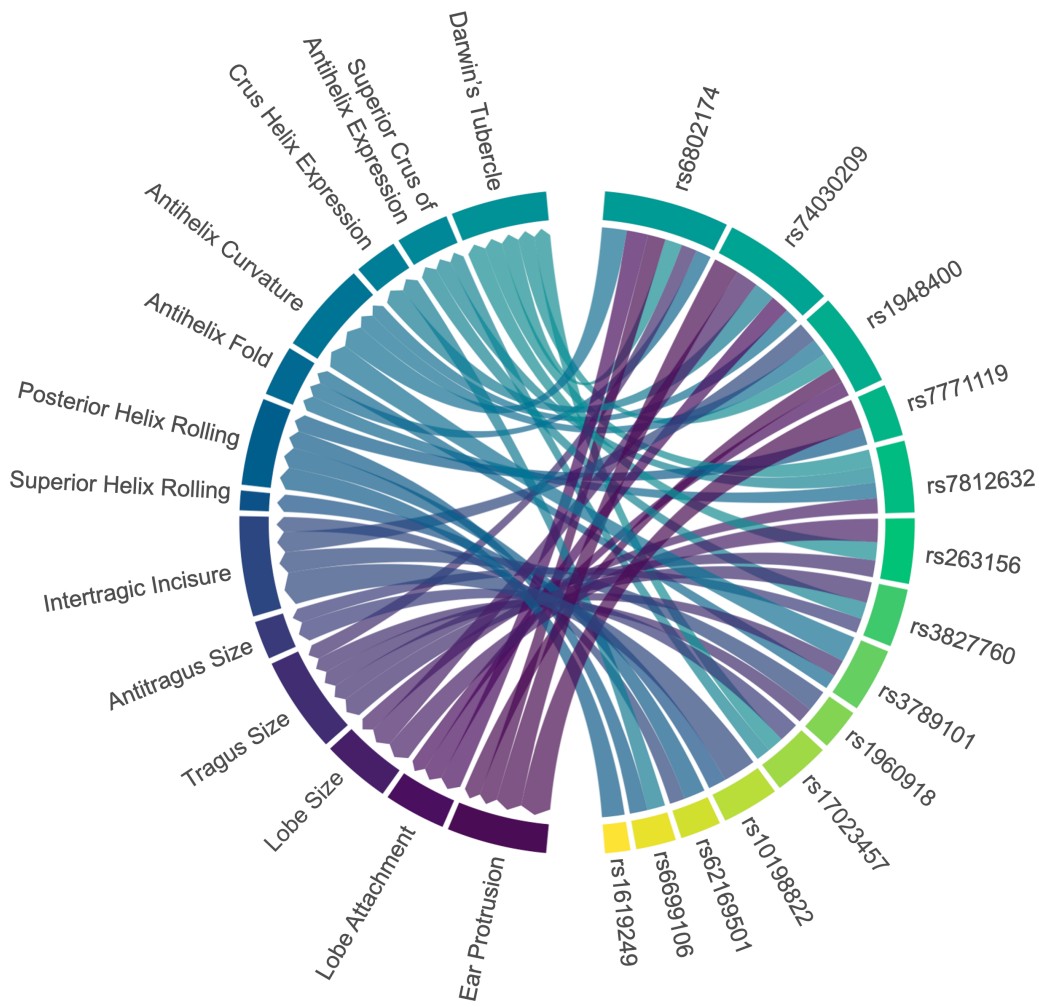

**Figure 3** **Associations between ear characteristics and SNP markers.** The left nodes represent ear traits, and the right nodes represent SNP markers. The connecting lines indicate significant associations ($p < 0.05$) between the different nodes. The thicker the line, the stronger the association intensity. SNP, single nucleotide polymorphism.

intertragic incisure, antihelix curvature, and Darwin's tubercle (Fig. 3). Due to the large number of ear traits included in our study, we have summarized the SNP marker that showed the strongest association with each trait, while the detailed information has been provided in Tables S7 to S20.

Four genetic variants (rs6802174, rs74030209, rs1948400, and rs7771119) were significantly associated with ear protrusion, with rs6802174 showing the strongest association. Compared to those with the CC genotype in rs6802174, individuals with the CG genotype had 3.09 times higher odds of having a large protrusion (level_2) (*p*-value = 3.03E−05). Furthermore, individuals with the GG genotype had 2.36 times higher odds of having a large protrusion (*p*-value = 0.0468).

Three SNPs (rs6802174, rs74030209, and rs1948400) were significantly associated with lobe attachment. Compared to those with the CC genotype in rs6802174, individuals with the GG genotype had a 2.70-fold increased likelihood ($p$-value $= 0.0122$) of having average lobe attachment (level_1). Three SNPs (rs74030209, rs7812632, and rs263156) explained the variation in lobe size. Compared to individuals with the CC genotype in rs74030209, those with the CT genotype have a significantly lower likelihood of having large lobe size, with ORs of 0.48 ($p$-value $= 0.0176$) and 0.35 ($p$-value $= 0.0019$), respectively.

Five SNPs (rs6802174, rs3827760, rs263156, rs3789101, rs1960918) were significantly associated with tragus size. In rs3827760, individuals with the AA genotype were 0.02 times ($p$-value $= 0.0125$) less likely to have large tragus size (level_2) compared to those with the GG genotype. Two genetic variants, including rs17023457 and rs3827760, were observed to be associated with antitragus size. In rs17023457, individuals with the CC genotype had a 2.26-fold increased likelihood ($p$-value $= 0.0155$) of having an average antitragus size compared to those with the TT genotype.

Four genetic predictors (rs10198822, rs1948400, rs62169501, and rs1960918) were significantly associated with intertragic incisure. Individuals with the TC genotype in rs10198822 had a 0.44-fold decreased likelihood ($p$-value $= 0.0011$) of having horseshoe-shaped intertragic incisure (level_1) compared to those with the TT genotype. In contrast, only one SNP, rs10198822, was observed to be associated with superior helix rolling. Individuals with the TC genotype exhibited a notable increase in the likelihood of a higher degree of superior helix rolling compared to those with the TT genotype. The odds increased by 3.58 times ($p$-value $= 0.0443$) for partial folded superior helix rolling (level_1), and by 5.72 times ($p$-value $= 0.0057$) for over-folded superior helix rolling (level_2).

The statistical significance was obtained for four SNPs (rs6699106, rs7771119, rs62169501, and rs1619249), which explained the variation in posterior helix rolling. In rs7771119, the presence of allele A was associated with a decreased risk of having a higher degree of posterior helix rolling. In rs7771119, allele A was associated with decreased risk of high-degree posterior helix rolling. Compared to CC, CA had 0.43 times lower odds ($p$-value $= 0.0083$), and AA had odds 0.18 times those of CC ($p$-value $= 0.0149$). Three SNPs—rs6802174, rs7812632, and rs3789101—were associated with the antihelix fold. In rs7812632, individuals with the GG genotype were 0.27 times ($p$-value $= 0.0372$) less likely to have partial folded antihelix (level_1) compared to those with the CC genotype.

Four genetic predictors (rs6802174, rs74030209, rs1948400, and rs3789101) were significantly associated with antihelix curvature. In rs3789101, individuals with the CC genotype had a 0.27-fold decreased likelihood ($p$-value $= 0.0303$) of having a strong antihelix curvature (level_2) compared to those with the GG genotype. Two variants (rs74030209, rs6699106) were associated with crus helix expression. Compared to individuals with the CC genotype in rs6699106, those with the TT genotype had a significantly lower likelihood of having a higher degree of crus helix expression, with ORs of 0.38 ($p$-value $= 0.0135$) and 0.36 ($p$-value $= 0.0359$), respectively.

Three SNPs (rs17023457, rs3827760, and rs7812632) explained the variation in the superior crus of antihelix expression. Compared to those with the GG genotype in rs3827760, individuals with the GA genotype had an increased odds of having prominent
antihelix superior crus (level_2) by 2.64 times ($p$-value = 0.0309). Five SNPs (rs6802174, rs17023457, rs7812632, rs1948400, and rs263156) were significantly associated with Darwin's tubercle. In rs263156, individuals with the GT genotype had a 0.34-fold decreased likelihood ($p$-value = 0.0229) of having a prominent Darwin's tubercle (level_2) compared to those with the GG genotype.

## Prediction results

For the 13 ear traits, the micro-average AUC in the three-class prediction for ear phenotypes ranged between 0.50 and 0.60. The MLR and AdaBoost models exhibited better prediction accuracy, whereas the SVM and RF models showed reduced performance, and the KNN model performed the worst (Fig. 4). Notably, for absent tragus (level_0), the predicted AUC exceeded 0.70 in the MLR and AdaBoost models (Fig. 4). The indicators describing prediction accuracy for the five models were presented in Table S21.

For absent tragus (level_0), the AdaBoost model achieved a prediction accuracy with an AUC of 0.73, an NPV of 0.78, a PPV of 0.45, a sensitivity of 0.54, and a specificity of 0.71. In contrast, the MLR model demonstrated lower accuracy, with an AUC of 0.71, an NPV of 0.78, a PPV of 0.49, a sensitivity of 0.54, and a higher specificity of 0.75. For medium tragus size (level_1), the AdaBoost model demonstrated an AUC of 0.65, an NPV of 0.71, a PPV of 0.48, a sensitivity of 0.38, and a specificity of 0.76. Meanwhile, the MLR model showed a lower accuracy, with an AUC of 0.60, an NPV of 0.68, a PPV of 0.38, a specificity of 0.69, and a sensitivity of 0.35. When dealing with prominent tragus cases (level_2), the AdaBoost model demonstrated an AUC of 0.62, alongside an NPV of 0.73, a PPV of 0.47, a sensitivity of 0.47, and a specificity of 0.72. The MLR model showed comparable performance, with an AUC of 0.66, an NPV of 0.72, a PPV of 0.45, a sensitivity of 0.44, and a specificity of 0.72.

In order to reduce model complexity, the tragus size was further converted into binary classifications. Specifically, tragus size was categorized into present (level_1 and level_2) *versus* absent (level 0). Predictive models for the binary classification task were established using AdaBoost and binary LR (BLR) models. The AdaBoost model achieved an AUC of 0.73, with an NPV of 0.59, a PPV of 0.68, a sensitivity of 0.65, and a specificity of 0.62 (Table 2). Similarly, the BLR model exhibited an NPV of 0.63, a PPV of 0.70, a specificity of 0.67, a sensitivity of 0.67, and a lower AUC of 0.71 (Table 2).

MDR analysis revealed a redundant interaction for the absent tragus (level_0) between rs3827760 in *EDAR* and rs1960918 in *LRBA*, which was indicated by blue lines in Fig. S3. When incorporated into the AdaBoost model, this interaction led to an improvement in the AUC to 0.74 ($\Delta$AUC = 0.0061) (Fig. 5A). Similarly, incorporating the interaction into the BLR model resulted in an AUC increase to 0.72 ($\Delta$AUC = 0.0172) (Fig. 5B). The confusion matrices and precision–recall curves of BLR and AdaBoost were displayed in Figs. S4 and S5, respectively. For both models, prediction accuracy parameters improved with the increase of the probability threshold (Table S22). For the AdaBoost model, setting the threshold at $p > 0.65$ led to high sensitivity, specificity, and PPV, while keeping the overall prediction error relatively low. However, at higher thresholds, the AdaBoost model tended to adopt a more conservative approach, resulting in a greater number of samples

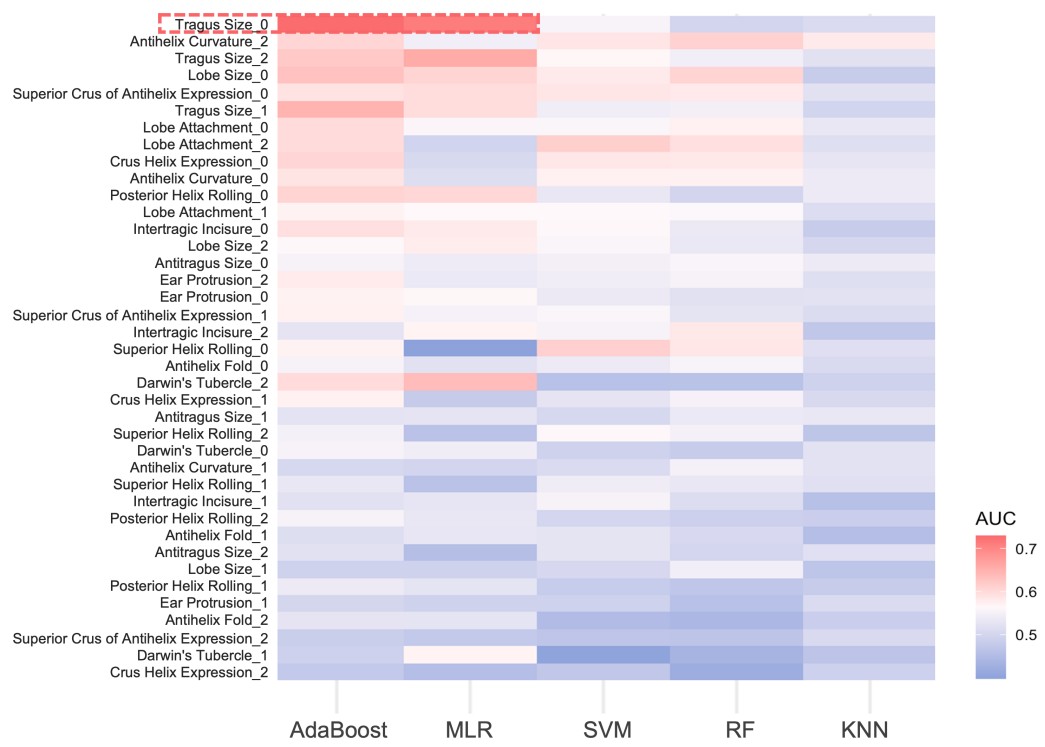

**Figure 4** **AUC heatmap for three classifications of ear traits.** The $y$-axis depicts the three subclasses under each ear trait category, while the $x$-axis indicates the five prediction models. Red signifies higher AUC values, indicating better predictive performance of the model on that particular classification trait. Blue, on the other hand, represents lower AUC values. The red frame highlights the traits that perform well in the AdaBoost and MLR models, with AUC values exceeding 0.7. MLR, multinomial logistic regression. SVM, support vector machines. RF, random forest. KNN, k-nearest neighbors. AUC, area under the curve.

**Table 2** **Prediction performance for absent tragus (level_0).**

| Model | Model | PPV | NPV | Sensitivity | Specificity | AUC |
|---|---|---|---|---|---|---|
| AdaBoost | SNPs | 0.68 | 0.59 | 0.65 | 0.62 | 0.73 |
| | SNPs + gene interactions | 0.71 | 0.60 | 0.64 | 0.65 | 0.74 |
| LR | SNPs | 0.70 | 0.63 | 0.67 | 0.64 | 0.71 |
| | SNPs + gene interactions | 0.71 | 0.65 | 0.71 | 0.63 | 0.72 |

**Notes.**
LR, logistic regression; SNP, single nucleotide polymorphism; AUC, area under the curve; PPV, positive predictive value; NPV, negative predictive value.

being classified as uncertain. In contrast, the BLR model exhibited more stable performance across different thresholds. Specifically, when the threshold was set to $p > 0.70$, the BLR model not only achieved high specificity, sensitivity, and PPV but also struck a good balance between accuracy and the ability to handle uncertain cases.

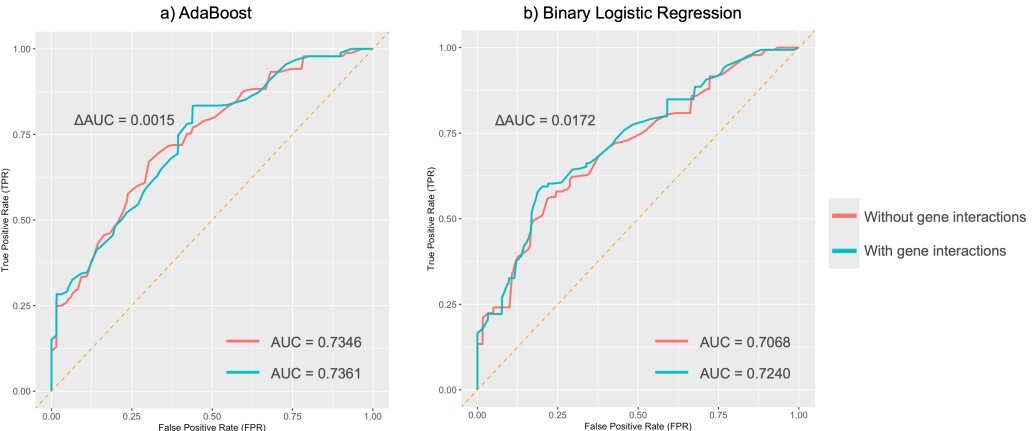

**Figure 5** **Average ROC curves for absent tragus (level_0).** (A) ROC curves of the AdaBoost model; (B) ROC curves of the binary logistic regression model. ROC, receiver operating characteristic. AUC, area under the curve.

## DISCUSSION

The external human ear is considered to be a polymorphic and polygenic structure with individual uniqueness that serves as an essential target in FDP studies. This research explored the associations between genetic markers and ear traits in 675 Chinese individuals and developed five ML models to evaluate the predictive efficacy of those markers. Among the five models, AdaBoost and MLR demonstrated comparable and better performance in predicting ear traits. In binary classifications, the AdaBoost and BLR models achieved medium predictive performance for the absent tragus (level_0), with AUC values exceeding 0.7. The inclusion of genetic interactions in our prediction models slightly enhanced the prediction capacity.

The human external ear, as a complex and integrated structure, exhibits varying degrees of interdependence among its traits. These interdependencies likely originate from the spatiotemporal regulatory mechanisms during embryonic development, which coordinate the morphology, sizes, and positions of different ear components (*Giraldez & Fritzsch, 2007*; *Anthwal & Thompson, 2016*; *Helwany, Arbor & Tadi, 2024*). In our work, we demonstrated the presence of correlations between ear traits. Previous studies have reported similar patterns of ear trait correlations in diverse populations (*Rani, Krishan & Kanchan, 2022*; *Adhikari et al., 2015*; *Li et al., 2023*). These comparable findings indicate that high correlations between various ear traits might be influenced by genetic variants with multi-trait effects (*Noreen et al., 2023*). Consequently, we chose to use the same SNPs to investigate multiple ear phenotypes, aiming to elucidate the underlying genetic associations.

Several SNPs deviated from HWE in our study. This result might be attributed to historical multi-ethnic admixture and migration patterns in Southwest China, which disrupted random mating assumptions. Additionally, our sample size was smaller than GWAS, further limiting statistical power for HWE testing (*Adhikari et al., 2015*; *Shaffer*

*et al., 2017*; *Wang et al., 2022*; *Li et al., 2023*). Notably, genetic loci flanking HWE-deviant SNPs, such as SP5, MRPS22, and EDAR, demonstrated significant associations with ear morphology in East Asian populations in GWAS (*Shaffer et al., 2017*; *Wang et al., 2022*; *Li et al., 2023*). Given our primary objective of developing predictive models, we retained these high-effect markers, consistent with *Noreen et al.*'s (*2023*) methodology for ear phenotype prediction.

In our study on the Chinese population, we verified significant associations between several SNP markers and ear traits. Specifically, we found the following associations: rs17023457 with antitragus size (*Adhikari et al., 2015*), rs74030209 (*Wang et al., 2022*), rs6802174, and rs1948400 (*Shaffer et al., 2017*) with lobe attachment, rs17023457 with the expression of the superior crus of antihelix (*Noreen et al., 2023*), rs7812632 (*Li et al., 2023*) and rs263156 (*Adhikari et al., 2015*) with lobe size, rs3789101 with folding of the antihelix (*Li et al., 2023*), rs263156 with tragus size, and rs1619249 (*Adhikari et al., 2015*) with posterior helix rolling. All of these associations were consistent with previous studies' findings. This consistency reinforces the credibility and generalizability of these associations across diverse populations. For several genetic markers, inconsistencies were observed between the current study and previously reported associations with ear phenotypes (*Adhikari et al., 2015*; *Shaffer et al., 2017*; *Wang et al., 2022*; *Li et al., 2023*; *Noreen et al., 2023*). These may stem from differences in study samples or methodology, leading to variations in association results.

In exploring the predictive power of genetic markers for ear phenotypes, our results showed that the average AUC values for predictions based on 13 ear features ranged from 0.50 to 0.60. This indicated that, given the current dataset, the predictive performance of these features is relatively low. Notably, *Noreen et al.*'s (*2023*) conducted a predictive study on 11 ear phenotypes among 300 individuals, reporting AUC values generally exceeding 0.77, with the highest reaching a remarkable level of 0.96. These significant findings offer valuable guidance for research in related fields; however, considering the impact of sample size on the stability of prediction results is essential. Compared to the study by *Noreen et al. (2023)*, our research incorporated a larger sample size, enabling the detection of more subtle genetic variations, thereby statistically enhancing the reliability and robustness of the findings.

In our study, an additional exploration was conducted to evaluate the predictive performance of various classifiers for ear phenotypes. According to the results obtained from five models—MLR, KNN, RF, SVM, and AdaBoost, AdaBoost and MLR models exhibited significant advantages. Additionally, A prevalent phenomenon was that the model exhibited better predictive performance for extreme categories (level_0 or level_2) compared to the intermediate categories (level_1) (Fig. 4, Table S21). This finding is in accord with previous research on EVC predictions (*Walsh et al., 2011*; *Walsh et al., 2013*; *Chaitanya et al., 2018*; *Kukla-Bartoszek et al., 2019*; *Katsara et al., 2021*; *Walsh et al., 2017*). The lower prediction accuracy for intermediate categories may stem from imprecise phenotype categorization and the fact that certain genetic variants remain unidentified (*Kukla-Bartoszek et al., 2019*; *Katsara et al., 2021*; *Liu et al., 2009*).

In binary classification tasks, both AdaBoost and BLR models achieved moderate predictive performance for absent tragus cases (level_0), with AUC values exceeding 0.7. Incorporating these interactions into BLR and SVM models using MDR led to an improvement in their performance. Notably, the AdaBoost model, including genetic interactions, achieved the highest AUC of 0.74. This further highlights the complex nature of the ear's unique morphology and underscores the significance of considering gene-gene interactions when studying ear morphology.

This study has several limitations that should be disclosed to facilitate future improvement. Given the variations in gene frequencies across sub-populations, GWAS commonly include sub-population information as covariates to prevent false-positive results. In our study, we recruited the Han population in Southwest China to reduce the impact of population stratification. However, due to the limited number of genetic markers detected, conducting population structure analyses is challenging. Therefore, future studies are expected to increase the number of SNPs using microarray or sequencing technologies, enabling more rigorous genetic analyses. Additionally, conclusions derived from this single-origin cohort require validation in external cohorts to establish generalizability.

Our study employed ear traits reported by prior GWAS to evaluate the predictive capacity of associated SNPs for these well-established phenotypes (*Adhikari et al., 2015*; *Shaffer et al., 2017*; *Wang et al., 2022*). While moderate predictive performance was achieved for absent tragus (AUC > 0.7), most traits demonstrated limited discriminative power with AUC values below 0.6 (Table S21). The superior NPV and specificity relative to PPV and sensitivity indicated stronger exclusionary capability in these models. However, when compared to highly predictable traits, including eye color, hair color, and skin color, current ear morphology prediction models remain unsuitable for direct forensic or clinical implementation due to the relatively low accuracy (*Walsh et al., 2011*; *Walsh et al., 2013*; *Chaitanya et al., 2018*). This limitation stems from constrained SNP coverage relative to the polygenic architecture of ear phenotypes. Future research is anticipated to extract quantitative geometric features, texture features, or high-resolution 3D features with high-density SNP analysis to advance ear phenotype prediction.

Research should also focus on the ethical issues of FDP in forensic identification. Non-visible traits, such as biogeographic ancestry, raise privacy concerns. Additionally, potential overlaps between phenotypic prediction and disease variants pose a risk of accidental disclosure of health information, leading to social discrimination. Moreover, striking a balance between societal gains and ethical protections is vital. In such a case, if the public prioritizes solving crimes over discrimination risks, it could lead to the use of FDP for predicting disease traits (*Kayser, 2015*). As an EVC, the external ear is recorded in identification systems, thereby minimizing privacy-related concerns. GWAS targeting SNPs associated with natural morphological diversity reduces the risk of disease-related information disclosure. Moreover, given the limited prediction accuracy, the utilization of ear phenotype prediction for disease prediction is restricted. With the increase in the number of SNPs and advancements in modeling algorithms, it is anticipated that regulatory frameworks for ear traits will be formulated to achieve a balance between forensic utility and ethical risks.

## CONCLUSIONS

In summary, our study is the first to explore the efficacy of multiple DNA-based predictive models for ear morphology in the Chinese population. We revalidated several SNP markers for their consistent associations with specific ear phenotypes reported in previous studies, and concurrently discovered associations between SNPs and different ear phenotypes. Our models demonstrated moderate predictive performance for absent tragus cases (level_0), even though the overall AUC for predicting ear traits was below 0.60. For absent tragus (level_0) predictions, both the AdaBoost model and the BLR model, after incorporating genetic interactions, achieved an AUC of 0.74 and 0.72, respectively. These findings underscore the potential practical applications of these SNPs and prediction models in forensic genetics and anthropology. Future research is expected to explore additional genetic markers and use alternative methods, such as polygenic risk scoring and deep learning, to enhance prediction performance. Furthermore, investigating the association between ear phenotypes and demographic factors, including ancestry, age, sex, and lifestyles, will offer critical insights into ear morphogenesis.

## ACKNOWLEDGEMENTS

We thank the support of the participants in this study. We thank the Editor and reviewers for their insightful and constructive comments.

### Funding

This work was supported by the Natural Science Foundation of China (grant number 82371897). The funders had no role in study design, data collection and analysis, decision to publish, or preparation of the manuscript.

### Grant Disclosures

The following grant information was disclosed by the authors:
The Natural Science Foundation of China: 82371897.

### Competing Interests

The authors declare there are no competing interests.

### Author Contributions

- Xindi Wang conceived and designed the experiments, performed the experiments, analyzed the data, prepared figures and/or tables, authored or reviewed drafts of the article, and approved the final draft.
- Zibo Zhao conceived and designed the experiments, performed the experiments, analyzed the data, prepared figures and/or tables, authored or reviewed drafts of the article, and approved the final draft.
- Jingting Wu performed the experiments, authored or reviewed drafts of the article, and approved the final draft.

- Yuan Li conceived and designed the experiments, prepared figures and/or tables, and approved the final draft.
- Yufei Yang performed the experiments, authored or reviewed drafts of the article, and approved the final draft.
- Bo Liu analyzed the data, prepared figures and/or tables, and approved the final draft.
- Chengye Zhou analyzed the data, prepared figures and/or tables, and approved the final draft.
- Chuanxu Wang analyzed the data, authored or reviewed drafts of the article, and approved the final draft.
- Xiaogang Chen analyzed the data, authored or reviewed drafts of the article, and approved the final draft.
- Feng Song conceived and designed the experiments, prepared figures and/or tables, authored or reviewed drafts of the article, and approved the final draft.

## Human Ethics

The following information was supplied relating to ethical approvals (*i.e.*, approving body and any reference numbers):

Sichuan University granted Ethical approval to carry out the study within its facilities (Ethical Application Ref: k2020032).

## Data Availability

The raw measurements are available in the Supplementary Files.

## Supplemental Information

Supplemental information for this article can be found online at http://dx.doi.org/10.7717/peerj.20169#supplemental-information.

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
