# Peer review of "DNA-based prediction of external ear morphology in the Chinese population: an exploratory study"

_PeerJ, doi:10.7717/peerj.20169_

## Round 0.1 · original submission · Major Revisions

· Academic Editor

Major Revisions

Reviewer 1 ·

Basic reporting

No comment

Experimental design

No mention of the inter-observer test in Methodology.
Before conducting Logistic regression analysis, it is essential to test the underlying assumptions. The assumptions are:
a) The observations must be independent of each other.
b) There should be no multicollinearity among the independent variables.
c) It requires the independent variables to be linearly related to a transformed version of the variables, i.e., to their log odds.

No attempt is made to test the assumptions.

Validity of the findings

No comment

Additional comments

No comment

·

Basic reporting

• "Inter-tragic" → should be corrected to "Intertragic" (for terminological consistency).
• The term "Helix rolling" carries anatomical ambiguity:
o If a natural structure is mentioned → "Helix curve",
o If surgical intervention is involved → "Helix fold" should be used.
• Instead of "folding of the anthelix", the terms "antihelical fold" or "antihelix fold", which are compatible with the literature, are recommended (Brito et al., Modified posterior triple scoring as a refinement technique in creating aesthetic antihelical fold in otoplasty, The American J Cosmetic Surgery, 2025;42(2); Khalid et al., Study of morphological variations of the human ear for its applications in personal identification, Indus J Bioscie Res, 2025;3(5)).
• "Facial attribute" should be replaced with "Facial features".

Experimental design

Ear morphology has become a more frequently studied topic in recent years. Approaching this topic from a genetic perspective has brought a very innovative perspective. In this context, the topic has been adequately stated in the Introduction. Perhaps something can be added to the purpose section about what kind of contribution it is expected to make to the field (industrially and/or technologically).

Validity of the findings

Problem and solution are clearly stated.
• Ethical permissions have been obtained.
• Additional details (e.g., PCR conditions, SNP selection criteria) can be provided in the Methods section for the reproducibility of the genetic analysis protocols used.

Additional comments

The manuscript presents a novel genetic approach to ear morphology, addressing a timely and increasingly researched topic. The ethical approval and clear problem definition are notable strengths. However, terminological inconsistencies and the need for a sharper focus on practical/industrial implications slightly diminish its impact. With minor revisions—particularly in terminology and methodological clarity—the study could become a valuable contribution to the field.
While I have evaluated the anatomical and methodological aspects of this study rigorously, my expertise in genetics is limited. I recommend that the genetic analyses reviewed by a specialist in this field to ensure the validity of these findings. The authors may also consider clarifying the genetic terminology/assumptions for a broader readership

Reviewer 3 ·

Basic reporting

The manuscript is written in clear, professional English. Minor grammatical issues and inconsistent terminology exist (e.g., switching between "LR" and "MLR" for logistic regression), but they do not impede comprehension.

The introduction provides a thorough overview of the forensic DNA phenotyping (FDP) field and situates the study well within current literature. However, the authors could improve clarity by summarizing key gaps in previous work more succinctly.

Experimental design

1. The manuscript focuses solely on 675 Chinese participants. It is unclear whether regional and ethnic subpopulations within China were adequately represented to account for population stratification. The authors should discuss the potential bias introduced by sampling and consider whether stratification or population structure analysis was performed.

2. Manual scoring of 13 ear traits is based on existing literature. However, trait categorization into three ordinal levels may introduce subjectivity. Although ICCs are reported, inter-rater reliability across multiple raters would strengthen confidence.

3. SNP selection was informed by prior GWAS and refined by allele frequency and LD filtering. The approach is sound, but inclusion of markers deviating from Hardy-Weinberg Equilibrium (8 out of 15 SNPs) should be justified more thoroughly.

4. The reported AUC values for most ear traits are modest (0.50–0.60) except for absent tragus (AUC ~0.7). The authors should discuss whether these predictive values are sufficient for forensic applications and provide justification for focusing on traits with low predictive power. Moreover, the limited PPV (~0.45–0.49) suggests poor clinical utility. This limitation should be highlighted.

5. While multiple machine learning models are tested (MLR, AdaBoost, SVM, RF, KNN), there is little discussion on why AdaBoost and MLR performed better. Did the authors perform hyperparameter tuning or evaluate model robustness on independent datasets? The manuscript would benefit from more rigorous model comparison using external validation cohorts.

6. The use of MDR to detect SNP-SNP interactions is innovative, but the biological significance of these interactions is not discussed.

7. Five models were compared using 10-fold cross-validation. The results indicate moderate performance (AUC \~0.50–0.60 for most traits), with best performance for "absent tragus" (AUC \~0.74). While this is promising, the overall predictive capacity remains modest.

8. Reducing trait categories to binary improves model performance slightly, particularly with AdaBoost. However, model gains are incremental and need stronger statistical justification. Confusion matrices and precision-recall curves could further clarify classification quality.

9. The authors acknowledge sample limitations and relatively low prediction accuracy for most traits. However, the discussion could be improved by clearly stating that the models are not currently suitable for forensic casework due to low trait-specific accuracy.

10. 10 analysis of sequences at the tools used for analysis is completely missing from the manuscript. Add the detailed sequence analysis with tools and methods.

11. The primer details have not been shared by the authors, leading to non-reproducibility.

12. Ethical considerations of applying FDP for forensic identification are underexplored and should be more fully addressed.
13. Workflow as illustrated properly. It is advisable to create a flowchart instead.

Validity of the findings

-

Additional comments

It is strictly recommended that the manuscript for major revision including all these points to taken care of.

---

## Round 0.2 · accepted · Accept

· Academic Editor

Accept

Of the three original reviewers, only one has agreed to re-review the revised manuscript. Thus, I confirmed that the authors addressed all points reasonably. Although this work still has many limitations, which are explained in the manuscript, I believe it is significant and acceptable as a pioneer in this field.

·

Basic reporting

Terminological arrangements have been completed, and a sufficient and valid explanation has been made regarding the use of the term "Helix rolling."

Experimental design

The requested statement was added to the Introduction by the authors.

Validity of the findings

Adequate explanations of additional information regarding the devices have been provided by the authors.

Additional comments

I would like to thank the authors for their diligent work and positive response to the comments.